# Is There a Place for PD-1-PD-L Blockade in Acute Myeloid Leukemia?

**DOI:** 10.3390/ph14040288

**Published:** 2021-03-24

**Authors:** Laura Jimbu, Oana Mesaros, Cristian Popescu, Alexandra Neaga, Iulia Berceanu, Delia Dima, Mihaela Gaman, Mihnea Zdrenghea

**Affiliations:** 1Department of Hematology, Iuliu Hatieganu University of Medicine and Pharmacy, 8 Babes Str., 400012 Cluj-Napoca, Romania; mesaros.oana@umfcluj.ro (O.M.); popescu.cristian@umfcluj.ro (C.P.); neaga.alexandra@umfcluj.ro (A.N.); mzdrenghea@umfcluj.ro (M.Z.); 2Department of Hematology, Ion Chiricuta Oncology Institute, 34-36 Republicii Str., 400015 Cluj-Napoca, Romania; iuliaberceanu@yahoo.com (I.B.); deli_dima@yahoo.com (D.D.); 3Department of Infectious Diseases, County Emergency Hospital Alba Iulia, 20 Decebal Str., 510093 Alba-Iulia, Romania; 4Department of Hematology, Carol Davila University of Medicine and Pharmacy, 050474 Bucharest, Romania; mihaela_dervesteanu@yahoo.com

**Keywords:** acute myeloid leukemia, checkpoint inhibitors, immune system, cancer

## Abstract

Checkpoint inhibitors were a major breakthrough in the field of oncology. In September 2014, based on the KEYNOTE-001 study, the Food and Drug Administration (FDA) approved pembrolizumab, a programmed cell death protein 1 (PD-1) inhibitor, for advanced or unresectable melanoma. Up until now, seven PD-1/PD-ligand(L)-1 inhibitors are approved in various solid cancers and hundreds of clinical studies are currently ongoing. In hematology, PD-1 inhibitors nivolumab and pembrolizumab were approved for the treatment of relapsed/refractory (R/R) classic Hodgkin lymphoma, and later pembrolizumab was approved for R/R primary mediastinal large B-cell lymphoma. In acute myeloid leukemia (AML), the combination of hypomethylating agents and PD-1/PD-L1 inhibitors has shown promising results, worth of further investigation, while other combinations or single agent therapy have disappointing results. On the other hand, rather than in first line, these therapies could be useful in the consolidation or maintenance setting, for achieving minimal residual disease negativity. Furthermore, an interesting application could be the use of PD-1/PD-L1 inhibitors in the post allogeneic hematopoietic stem cell transplantation relapse. There are several reasons why checkpoint inhibitors are not very effective in treating AML, including the characteristics of the disease (systemic, rapidly progressive, and high tumor burden disease), low mutational burden, and dysregulation of the immune system. We here review the results of PD-1/PD-L1 inhibition in AML and discuss their potential future in the management of this disease.

## 1. Introduction

The immune system has a complex role in defending the host against infections, against the growth of tumor cells, and also in tissue repair [1]. When an antigen is identified by the adaptive immune system, the antigen-presenting cells (APCs) become active and migrate to the lymph nodes in order to activate B and T cells. For the activation of the T cells, two signals are needed. The first signal requires the interaction between the T cell receptor (TCR) situated on the T cells, and the epitope of the antigen, presented together with the MHC (major histocompatibility complex) molecules, situated on the APCs. The second, co-stimulatory, signal represents the interaction between CD28 (on the T cells) and B7.1 (CD80) and respectively B7.2 (CD86) (on the APCs) [2]. These two signals promote the proliferation, differentiation, and survival of T cells. In addition to the aforementioned co-stimulatory signaling, co-inhibitory signals also exist [3]. There are several molecules that control the response of the immune system and downregulate T cell activation, called immune checkpoints. The most investigated immune checkpoints are: cytotoxic T-lymphocyte-associated protein 4 (CTLA4) [4], programmed cell death protein 1 (PD-1) [5], T cell immunoglobulin-3 (TIM-3), lymphocyte activation gene-3 (LAG-3) [6], B and T lymphocyte attenuator (BTLA) [7], V-domain Ig suppressor of T cell activation (VISTA) [8], and T cell immunoglobulin and ITIM domain (TIGIT) [9]. Several studies revealed that tumor cells use these pathways to escape the immune system and to disseminate [10]. These discoveries led to the development of novel agents—checkpoint inhibitors, which “release the brakes” of the immune system.

## 2. PD-1, PD-L1, and PD-L2 Biology

PD-1, also called CD279, is a glycoprotein cell receptor which is part of the superfamily of B7-CD28 and it is encoded by a gene (pdcd-1) situated on chromosome 2 (2q37) [11]. Pdcd-1 consists of five exons [12]. Exon 1 encodes an extracellular peptide, exon 2 an immunoglobulin variable domain, exon 3 a transmembrane domain, and exons 4 and 5 encode an intracellular domain. PD-1 is a 50–55 kDA protein composed of 288 amino acids [13,14].

PD-1 cDNA was first isolated in 1992 [15]. Its role in regulating the response of the immune system has been proved by PD-1 negative mouse models, which developed different autoimmune diseases [16,17]. Under physiological conditions, the PD-1-PD-ligand(L)1 pathway protects against autoimmunity, promoting apoptosis of effector T cells and stimulating the development of regulatory T cells (Tregs) from naïve T cells. Tregs are a subtype of T cells involved in maintaining peripheral tolerance, by downregulating effector T cells. They express CD4, CD25, and FOXP3 [18].

PD-1 is expressed on B cells, natural killer cell (NK cells), CD4+ T cells, CD8+ T cells, CD4- CD8- T cells, activated monocytes, dendritic cells (DC), macrophages, and immature Langerhans cells [11]. Its expression is enhanced by IL-2, IL-21, IL-15, IL-7, type 1 interferons (IFNs), IL-6, and IL-12 [11,19]. PD-1 has two known ligands: PD-L1 and PD-L2.

PD-L1 (called CD274 or B7-H1) was discovered in 1999 [20] and is a type I transmembrane protein, composed of 290 amino acids. It has 33 kDa and it is composed of two extracellular domains (IgV- and IgC-like domains), one transmembrane and one intracellular domain. PD-L1 is encoded by the Cd274 gene on chromosome 9 (9p24) [12]. It is expressed on lymphoid tissue (T cells, B cells, macrophages, and DC) and also on non-lymphoid structures (vascular endothelial cell, beta cells in the pancreas, placenta, and testicle) [11,21]. PD-L1 is expressed on cells infected by viruses such as Ebola virus, friend retrovirus, human immunodeficiency virus, herpes simplex virus type 1, hantavirus, influenza A virus, Japanese encephalitis virus, Kaposi’s sarcoma-associated herpesvirus, lymphocytic choriomeningitis virus, respiratory syncytial virus, and varicella zoster virus [22]. Furthermore, PD-L1 is overexpressed in several types of cancers and hematological malignancies such as colorectal, ovarian, pancreatic, gastric, renal, breast, lung, thyroid, testicular cancer, melanoma, and Hodgkin lymphoma (HL) [23,24]. PD-L1 overexpression is upregulated by interferon gamma through the JAK-signal transducer and activator of transcription (STAT) pathway [25]. Other studies have shown PD-L1 overexpression in diffuse large B cell lymphoma (DLBCL), follicular lymphoma, peripheral T-cell lymphoma [26,27], multiple myeloma (MM) [28], chronic lymphocytic leukemia (CLL) [29], and acute myeloid leukemia (AML) [30].

PD-L2 (called CD273 or B7-DC) is also a type I transmembrane receptor, composed of 273 amino acids, and encoded by Pdcd1lg2 gene, situated on chromosome 9. It is expressed on macrophages, DC, and mast cells [12]. PD-L2 is associated with a higher production of T-cells and IFN gamma [31]. However, its role and mechanism of action is not completely understood.

## 3. PD-1 and PD-L1 Inhibitors in Cancer

The advent of PD-1 and PD-L1 inhibitors was a major breakthrough in the treatment of several solid cancers. One of the first clinical studies, in 2010, showed the efficacy of MDX-1106 (nivolumab), a PD-1 inhibitor. Thirty-nine patients with refractory metastatic melanoma, renal cell carcinoma, non-small-cell lung cancer (NSCLC), prostate cancer, or colorectal cancer (CRC) were included in the study. Out of the 39 patients, one achieved complete remission (CR), and two partial remissions (PR) [32]. Later, pembrolizumab, a humanized monoclonal antibody which blocks the interaction between PD-1 and PD-L1, was approved in 2015 for unresectable or metastatic melanoma and in 2016 for metastatic NSCLC and head and neck squamous carcinoma (HNSCC) with progression on or after platinum-containing therapy. In October 2016, pembrolizumab received approval for the treatment of patients with metastatic NSCLC which express PD-L1, and who have progressed on or after platinum-containing treatment. In 2017, pembrolizumab was approved for the treatment of refractory classic HL (cHL) [33]. Later, the CkeckMate 227 trial showed better results (prolonged overall survival (OS), better response rate) in the treatment of NSCLC with the combination of nivolumab and ipilimumab (anti-CTLA-4 antibody) [34].

Atezolizumab is a humanized monoclonal antibody which targets PD-L1, and inhibits the interaction between PD-1 and its ligand [35]. Phase II and phase III trials showed efficacy of atezolizumab (MPDL3280A) in invasive bladder cancer [36], in platinum treated patients with locally advanced/metastatic urothelial carcinoma [37], in triple negative advanced/metastatic breast cancer [38], and in NSCLC [35].

Cemiplimab is a human monoclonal antibody targeting PD-1. Based on EMPOWER-CSCC 1 trial, cemiplimab was approved by FDA in 2018 and by EMA in 2019, for the treatment of locally advanced/metastatic cutaneous squamous cell carcinoma not eligible for curative radiotherapy or surgery [39]. Several studies in other solid cancers and hematological malignancies are currently ongoing.

Durvalumab (MEDI4736) is a fully human monoclonal antibody targeting PD-L1 [40] which improved the OS of patients with advanced NSCLC [41], advanced urothelial bladder cancer [42], HNSCC progressed on platinum-based chemotherapy [43]. Several phase I and phase II studies showed efficacy in triple negative breast cancer [44] and in advanced-stage ovarian cancer in combination with tremelimumab [45] or olaparib [46]. Studies are ongoing.

Avelumab (MSB0010718C) is a fully human IgG1 anti-PD-L1 monoclonal antibody [47] which showed promising results in phase Ib clinical studies in patients with metastatic Merkel cell carcinoma [48], advanced unresectable mesothelioma [49], ovarian cancer [50], NSCLC [51], gastric or gastroesophageal junction cancer [52], refractory metastatic urothelial carcinoma [53], and renal cell carcinoma [54].

Spartalizumab (PDR001) is a humanized IgG4 anti PD-1 monoclonal antibody tested in various types of advanced or metastatic solid cancers [55,56]. A phase Ib open label study of spartalizumab and/or MBG453 and/or decitabine in patients with relapsed/refractory (R/R) AML or high-risk myelodysplastic syndrome (HR MDS) is currently recruiting [57].

Up until now, FDA approved PD-1 and PD-L1 checkpoint inhibitors for melanoma, renal cell carcinoma, head and neck cancer, urothelial carcinoma, CRC, hepatocellular carcinoma, small and non-small cell lung cancer, esophageal squamous cell carcinoma, cervical cancer, Merkel cell carcinoma, bladder cancer, certain types of endometrial carcinoma, and breast cancer. In hematologic malignancies PD-1, PD-L1 inhibitors are approved in HL and primary mediastinal large B-cell lymphoma (PMBCL).

Interestingly, a meta-analysis including 19 randomized clinical trials and 11,379 patients, showed that PD-1 inhibition leads to a higher OS and progression free survival (PFS) compared to PD-L1 inhibition (as single agent or in combination), in different types of cancers. While PD-1 inhibitors block the interaction between PD-1 and PD-L1 and, respectively, PD-L2, PD-L1 blockers inhibit only the PD-1/PD-L1 axis, allowing the tumor cells to escape the immune system via PD-1-PD-L2 axis and thus explaining the abovementioned results [58].

A high tumor mutational burden (TMB) can increase the diversity of tumor cell antigens, and will increase the chance that some of the antigens will be recognized by the immune system. It has been demonstrated that a high TMB is associated with a better response to PD-1 inhibition [59]. Moreover, as expected, a high expression of PD-L1 on tumor cells is associated with better response to checkpoint inhibitors [59].

## 4. PD-1 and PD-L1 Inhibitors in AML

### 4.1. Introduction

AML is a heterogenous disease characterized by the proliferation of abnormal myeloblasts in the bone marrow. AML has a dismal prognosis in young patients capable of withstanding high doses of chemotherapy and even worse in elderly, frail patients with comorbidities. Even though the understanding of AML pathogenesis has improved over the last decades, the standard treatment for AML patients dates back to 1973 [60]. The discovery of t(15;17) and its personalized treatment, FLT3-inhibitors, BCL2-inhibitors, monoclonal antibodies, epigenetic regulators, and bispecific T-cell engager (BiTE) antibodies improved the OS of these patients [61]. The five year OS in young patients increased from 13% to 49% and in elderly patients from 8% to 13% from 1970 to 2015 [62]. Even so, there is an urgent need of novel, personalized drugs.

### 4.2. Immune Checkpoint Blockade in AML—Why Was It Bound to Fail?

Checkpoint inhibitors were a major breakthrough in the treatment of solid cancers, especially in those with high mutational burden due to the higher amount of neo-antigens [63]. This paradigm was later applied to hematological malignancies but with less success possibly due to different immune pathways and a higher immune tolerance [64]. However, as expected, in cHL, PD-1 inhibitors are of great benefit and demonstrated excellent results [65,66].

Compared with cHL, AML has different characteristics. It is an aggressive, rapid progressive disease, which does not allow the immune system to develop a proper anti-leukemic response. A study in a murine model showed that localized implantation (subcutaneous) of leukemic cells triggers a response from the immune system, as opposed to a systemic (intravenous) route, which generates a tolerant state towards the malignant cells [67]. Moreover, the high tumor burden affects the response to PD-1 inhibitors [68]. Furthermore, AML has a low mutational burden and the newly formed antigens are expressed in different other tissues of the host [69]. Interestingly, some case series were reported, which describe spontaneous remissions in patients with AML, especially after an immune event (e.g., infections) suggesting the importance of the immune response [70]. Several studies suggested that Tregs are increased, both in peripheral blood and bone marrow of patients with AML, compared to healthy participants. However, there are conflicting results regarding the significance of increased Tregs [71]. Wang et al. demonstrated with their mouse model that the accumulation of Tregs in the leukemic microenvironment has a dismal prognosis. Interestingly, the destruction of Tregs in the tumor environment inhibits the anti-leukemic immune response. Thus, blocking the accumulation of Tregs in the tumor environment would be an attractive therapeutic approach. Wang et al. also demonstrated that a low number of Tregs increased survival and decreased leukemia burden [72]. Careful consideration must be given to the depletion of Tregs, which could lead to severe autoimmune events. Gutierrez et al. demonstrated in in vivo and in vitro studies that midostaurin, a FLT3 inhibitor, decreases Tregs in AML patients and healthy participants suggesting that a combination with immunotherapy could be possible and needs further investigation [73]. Other factors that promote tumor evasion and suppression are exhausted T cells, decreased function of T helper cells and production of cytokines and enzymes that suppress the immune system (e.g., indoleamine 2,3-deoxygenase 1, L-kynurenine, and 2-hydroxyglutarate) [71]. Another way to escape the immune system is for the AML cells to downregulate the MHC class II, a phenomenon especially seen in the post-transplantation setting [74].

In AML, the overexpression of CD47 inhibits phagocytosis via signal receptor protein-alfa (SIRP-alfa) on macrophages and DC and is associated with poor survival [75,76]. These discoveries led to the development of CD47 checkpoint inhibitors which are currently examined in several studies, for different types of cancer [77].

In conclusion, AML cells develop several mechanisms for hijacking the immune system via the immune checkpoints (Figure 1).

### 4.3. Immune Checkpoint Blockade in AML—Why Was It Bound to Succeed?

A trial which included 124 bone marrow biopsies from patients with MDS, AML and chronic myelomonocytic leukemia (CMML) showed that PD-1, PD-L1, PD-L2 and CTLA4 were upregulated in CD34+ cells. AML and MDS bone marrow biopsies showed PD-1 positivity on the stroma and PD-L1 positivity on the blast population. Statistic correlations demonstrated that PD-1 expression is associated with increased age while PD-L2 expression is associated with female gender [78]. While there are several clues that PD-1 and PD-L1 inhibitors would lack the success seen in solid cancers, some combinations are worth further investigations. PD-1 expression on T cell is regulated by DNA methylation. Apparently, hypomethylating agents (HMAs) are able to upregulate PD-1 expression on T cells, thus creating a resistance mechanism [79]. A clinical trial testing HMA and vorinostat showed that upregulated PD-L1 and PD-L2 leads to a lower median survival as compared to patients without upregulated PD-L1 and PD-L2 (6.6 months vs. 11.7 months) [80]. These conclusions led to the development of several clinical trials that tested the combination between a HMA and PD-1/PD-L1 blockers.

### 4.4. Results in AML

Based on the Viale-A study, in USA, the standard of care for elderly patients with AML is now the combination of HMAs (azacitidine/decitabine) plus a BCL2 inhibitor (venetoclax). This combination improved OS (14.7 months vs. 9.6 months) and increased CR (36.7% vs. 17.9%) as compared with azacitidine alone [81]. Even with this combination elderly patients have a dismal prognosis. Thus, further investigation is needed. Currently, the combination of venetoclax + HMAs + pembrolizumab is evaluated to assess the percentage of patients who achieve undetectable minimal/measurable residual disease (MRD) compared to venetoclax + HMA (Blast MRD AML-2 study) [82].

PD-1, PD-L1 inhibitors are also studied in the R/R setting. A single arm, phase II clinical trial assessed the efficiency of azacitidine plus nivolumab in 70 elderly patients with R/R AML. The overall response (OR) was 33% (15 CR/complete remission with incomplete hematological recovery (CRi), 1 partial remission (PR), 7 hematological improvement, 9% had stable disease (SD), and 58% no response). Overall response rate (ORR) to HMAs was better in naïve patients [83]. Another clinical trial, which enrolled 10 patients, tested the efficacy of pembrolizumab and decitabine in R/R AML. At the end of the eighth cycle, four patients presented SD, four progressed, one was MRD negative, and one was excluded from the study due to toxicity, during the fifth cycle. Median OS was 7 months with a median time of follow-up of 13 months [84].

Resistance to PD-1 inhibitors may be due to up-regulated CTLA-4 [85]. This observation led to studies that assessed the combination of HMA + PD-1 inhibitor + CTLA-4 inhibitor. Daver et al. showed in a phase II trial that azacitidine + nivolumab + ipilimumab improved OS compared to azacitidine + nivolumab and azacitidine alone (10.5, 6.4, and 4.6 months, respectively) [83].

PD-1 inhibitors were also tested in combination with high dose chemotherapy. A single arm phase I–II clinical trial assessed nivolumab plus cytarabine and idarubicin in 44 patients with newly diagnosed AML or HR MDS. Median event free survival (EFS) was not reached at a median follow-up of 17.25 months. The median OS was 18.54 months [86]. Pembrolizumab was associated with high dose cytarabine in a clinical trial, which enrolled 37 patients with R/R AML. OR was 46% and CR was 38%. Nine patients received maintenance with pembrolizumab but seven of them relapsed [87].

There are several ongoing studies that investigate PD-1 inhibitors and MRD, and results are pending. Blast MRD AML-1 trial, which is currently ongoing, assesses the percentage of patients who achieve undetectable MRD with pembrolizumab in combination with intensive chemotherapy compared to chemotherapy alone. In a similar manner, the Blast MRD AML-2 trial assesses pembrolizumab in combination with azacitidine and venetoclax compared to azacitidine and venetoclax alone [88,89]. Moreover, nivolumab is tested as a single agent for eliminating MRD positivity in patients in complete remission [90]. Several authors consider PD-1/PD-L1 inhibitors a possible therapeutic approach in eliminating MRD positivity [68,91].

Allogenic hematopoietic stem cell transplantation (allo-HSCT) is an effective immunotherapy, for relapsed or high risk patients with AML, which uses the donor’s immune cells to develop a response towards the disease (graft versus tumor effect). Patients who relapse after allo-HSCT have a poor prognosis. In these conditions, treatment is channeled towards harnessing the immune system either with a second HSCT, in selected cases, or with donor lymphocytes infusion (DLI) [92]. Another option, available in the future could be the PD-1/PD-L1 blockade. Several studies suggest that PD-1/PD-L1 blockers are capable of inducing graft versus leukemia effect (GVL) [2,93]. Interestingly, adding PD-L1 inhibitors early after allo-HSCT triggers GVL but with high graft versus host disease (GVHD) in comparison to adding them later in the course of the treatment which is associated with GVL with no GVHD [93]. Pembrolizumab was assessed in the setting of relapsed hematological malignancies following allo-HSCT. Eleven patients were included in the study, eight with AML, two with DLBCL, and one with HL. Of the seven patients evaluable, three patients progressed, two had SD, and two achieved CR [94]. The combination of nivolumab and ipilimumab is currently investigated in post allo-HSCT relapse in patients with AML and MDS [95]. A case series reported three AML patients treated with PD-1 inhibitors in the post allo-HSCT setting from which one achieved CR, one SD and one progressed [96]. PR was achieved with nivolumab in a heavily treated patient with HL, relapsed after allo-HSCT with the expense of gastrointestinal and hepatic GVHD [97]. Furthermore, another case series comprised of 31 patients, mostly HL cases, reported 77% ORR with PD-1 inhibitors treatment in the aforementioned setting [98].

A meta-analysis of 24 articles evaluated the benefit of checkpoint inhibitors before or after allo-HSCT in different hematological malignancies and revealed that adding checkpoint blockers before or after allo-HSCT leads to higher rate of chronic, acute and hyperacute GVHD. T cells with low expression of PD-1 persist for 10 months or more leading to a higher risk of GVHD [99]. Several studies, especially in cHL, showed that PD-1 inhibitors are highly efficient in the relapsed setting, after allo-HSCT, at the cost of a higher rate of GVHD [96,98,100]. Thus, these agents could be used in relapsed AML but the risk GVHD flares should be thoughtfully considered [68]. On the other hand, Oran et al. demonstrated that the use of checkpoint inhibitors prior to allo-HSCT is feasible and post allo-HSCT administration of cyclophosphamide reduces the risk of acute GVHD [101].

After IFN-gamma exposure, PD-L1 expression had a minor increase in healthy patients but increased significantly in AML patients [102]. Several studies demonstrated that the expression of PD-L1 in AML blasts is restricted at diagnosis [102,103] and upregulated in relapse, progression, and CR [102]. The overexpression of PD-L1 in AML patients in CR is explained as a response of the malignant cells to chemotherapy (adaptive resistance) [102]. A study from 2018, conducted on 55 patients, demonstrated that the incidence of PD-L1 expression is higher in patients with leukocytosis [30]. Other studies showed a correlation between TP53 mutation and the overexpression of PD-L1 [2,104].

A 36 patient study demonstrated that PD-L1 expression level is a negative prognostic factor in patients with FLT3- ITD (internal tandem duplications) and concomitant NPM1 mutation [105]. A small study published in 2018 showed that NPM1 mutated blasts showed a higher expression of PD-L1 when compared to NPM1wild type AML blasts [106].

A review of ongoing or completed clinical trials of PD-1, PD-L1 inhibitors in AML is presented in Table 1.

## 5. Toxicities

Even though checkpoint inhibitors are not associated with the classic side effects of chemotherapy, they are not completely harmless, as they are associated with immune-related adverse events (irAEs). These adverse events can vary from asymptomatic to life threating or rarely even death and they can affect almost every organ or system at any time of the treatment. Depending on the severity of the adverse events the treatment may vary from monitoring to high dose of corticosteroids [132]. Table 2 represents the most common adverse events following the treatment with PD-1/PD-L1 inhibitors.

## 6. Conclusions

Checkpoint inhibitors were a major breakthrough in the treatment of solid cancers, and raised hope for a new, less aggressive therapy in hematological malignancies with suboptimal treatment results like AML. Despite promising results in some subtypes of lymphoma, currently being approved in HL and PMCBL, PD-1/PD-L1 checkpoint inhibitors have shown disappointing results in trials investigating their use as single agent, in both, de novo and relapsed AML. However, the combination of these agents with non- aggressive approaches, like HMAs, has shown promising activity in AML and is being currently studied in ongoing clinical trials. We believe that another potential use of PD-1/PD-L1 inhibitors in AML could be in the setting of either consolidation or maintenance where, in the presence of an at least partially restored immune system, they could promote MRD negativity. In this respect, they could be used either as single agents or in combinations. A very interesting therapeutic application, albeit of limited use, of checkpoint inhibitors in AML, could be in the post allo-HSCT setting, where, in the presence of AML relapse/progression, these agents might be useful in augmenting the immune reactivity of the graft, boosting the GVL effect, at the expense of also enhancing GVHD. To summarize, even though immune checkpoint blockade did not meet the high expectations they were credited with in AML, they are still a welcome addition to the limited therapeutic options in this group of diseases.

## Figures and Tables

**Figure 1 pharmaceuticals-14-00288-f001:**
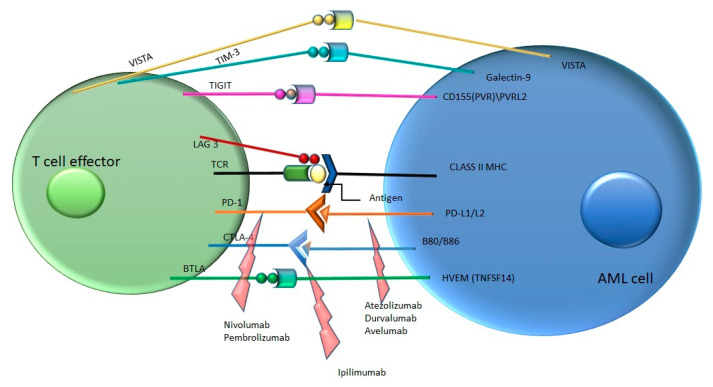
Immune checkpoint pathways and their inhibitors in acute myeloid leukemia (AML) (inhibitors—marked with red arrows). PVR—poliovirus receptor, PVRL2—poliovirus receptor-related 2, HVEM—herpesvirus entry mediator.

**Table 1 pharmaceuticals-14-00288-t001:** Completed and ongoing clinical trials of programmed cell death protein 1 (PD-1), PD-ligand(L)1 inhibitors in AML.

Disease Stage	Therapeutic Agents	Study Design	Participants	References
AML/HR MDS 18–60 years or >60 eligible for intense chemotherapy	cytarabine 1.5 g/m^2^ by 24 h continuous infusion daily on days 1–4 (3 days in patients > 60 years) and idarubicin 12 mg/m^2^ daily on days 1–3. nivolumab 3 mg/kg, day 24 every 2 weeks, 1 year for responders	Single-arm, phase II part of the phase I/II study	*n* = 44	[86]
R/R AML > 18 years	azacitidine iv/sc 75 mg/m^2^ days 1–7 + nivolumab iv 3 mg/kg days 1 and 14, every 4 to 6 weeks	Non-randomized, open-label, phase II study	*n* = 70	[83]
Newly diagnosed patients with TP53 mutated AML	Induction: nivolumab iv day 15 of cycle 1 and days 1 and 15 of subsequent cycles, decitabine 1–10 of induction cycle 1 and venetoclax orally daily on days 1–21 Maintenance: nivolumab iv: days 1 and 15, decitabine iv: days 1–5, and venetoclax po: days 1–21	Non-randomized, open-label, pilot study	*n* = 13	[107]
AML patients in first CR/CRi after intense chemotherapy not candidates for HSCT	nivolumab iv every 2 weeks for 46 cycles vs. clinical observation	Randomized, open-label, phase II study	*n* = 82	[90]
AML/HR MDS 18–60 years or >60 eligible for intense chemotherapy or R/R AML/MDS for phase I	Phase I: nivolumab iv 1 mg/kg on day 24 of a 28 days cycle and after cycle 2, nivolumab iv every 2 weeks, 1 year + idarubicin 12 mg/m^2^ IV days 1–3 + cytarabine iv 1.5 g/m^2^ days 1–4 + solumedrol 50 mg/dexamethasone iv 10 mg days with 1–4. Phase II: nivolumab maximum tolerated dose	Non-randomized, open label, phase I/II study	*n* = 75	[108]
R/R AML or MDS patients following allogenic HSCT	nivolumab iv, days 1 and 15 vs. ipilimumab iv day 1 vs. nivolumab iv, days 1, 14, and 28 + ipilimumab iv, day 1	Non-randomized, open label, phase I study	*n* = 55	[95]
AML patients ≥ 55–85 years, in first/second CR, suitable for haploidentical transplant	cytarabine iv 500–1000 mg/m^2^ bid days-2–4 + G-CSF, day 0 + nivolumab 40 mg, day 5 vs. cytarabine iv 500–1000 mg/m^2^ bid days 1–3 + nivolumab 40 mg day 1	Randomized, open-label, phase II study	*n* = 16	[109]
R/R AML/biphenotypic patients or newly diagnosed ≥ 65 years AML patients, unfit for in high dose chemotherapy	azacitidine iv/sc, days 1–7 or days 1–4 and 7–9 + nivolumab iv, days 1 and 14 (cycle 1–4) and day 1 (cycle 5 and subsequent) vs. same regimen + ipilimumab iv day 1 and then every 6–12 weeks	Non-randomized, open label, phase II study	*n* = 182	[110]
HR of relapse in AML patients in CR/CRi/CRp/PR	nivolumab iv, days 1 and 15. (cycles 1–5) and nivolumab iv, day 1, (cycle 6–12), and nivolumab iv, day 1(every 3 cycles starting from cycle 12) or continue nivolumab days 1 and 15 if progressive disease	Non-randomized, open label, phase II study	*n* = 30	[111]
R/R AML/HR-MDS, IDH1 mutated	ivosidenib PO 500 mg/day + nivolumab 480mg on day 1 cycle 2.	Non-randomized, open label, phase II study	*n* = 45	[112]
18–70 years AML/HR MDS eligible for HSCT	nivolumab iv (1 mg/kg or 3 mg/kg), 12 doses, day 1 every 3 weeks, 12 cycles vs. Ipilimumab (0.3 mg/kg/1.0 mg/kg/3.0 mg/kg), day 1, every 3 weeks, 6 cycles vs. nivolumab iv (3 mg/kg), 12 doses, day 1 every 3 weeks, 12 cycles + ipilimumab (0.3 mg/kg/0.6 mg/kg/1.0 mg/kg), day 1, every 3 weeks, 6 cycles	Non-randomized, open label, phase I study	*n* = 21	[113]
IPSS-1, IPSS-2, HR MDS, low blast count AML	DEC-205/NY-ESO-1 fusion protein CDX-1401 intracutaneously + poly ICLC sc, day-14 and day 15 (cycle 1–4), and day 1 of every 4 courses (cycle 5 and after) + nivolumab iv days 1 and 15 and decitabine iv, days 1–5	Non-randomized, open label, phase I study	*n* = 8	[114]
Recurrent AML/ALL/CLL/CML BCR-ABL+/HL/MM/non-Hodgkin Lymphoma/MDS/MPN/Other hematologic malignancies after allo-HSCT	Induction: ipilimumab iv, day 1+ nivolumab iv, day 1. (cycles of 21 days). Maintenance: ipilimumab iv every 12 weeks + nivolumab iv every 2 weeks in the absence of progressive disease or toxicity.	Non-randomized, open label, phase I/IB study	*n* = 71	[115]
HR AML in remission not eligible for HSCT	nivolumab 3 mg/kg iv every 2 weeks for 6 months. After 6 months nivolumab was given every 4 weeks until 12 months on the study, and every 3 months until relapse	Non-randomized, open label, phase II study	*n* = 8	[116]
R/R AML who have exhausted standard of care options	flotetuzumab in step-up dose, followed by continuous infusion flotetuzumab, starting at week 2 of cycle 1 and continuing through each 28-day cycle. MGA012 every two weeks.	Non-randomized phase I study		[117]
R/R AML	atezolizumab iv on day 22 of cycle 1 and on days 8 and 22 on subsequent cycles + Hu5F9-G4 1 mg/kg on days 1 and 4, 15 mg/kg on day 8, 30 mg/kg on day 11, and continue with 30 mg/kg every week	Non randomized, Open-label phase Ib study	*n* = 21	[118]
R/R or newly diagnosed patients with AML unfit for intensive chemotherapy	atezolizumab 840 mg iv on days 8 and 22 + guadecitabine 60 mg/m^2^ sc on Days 1–5	Non randomized, open-label phase Ib study	*n* = 40	[119]
≥60 years AML patients in CR/CRi, MRD+ not eligible for HSCT	BL-8040 SC 1.25 mg/kg days 1–3 of each cycle + atezolizumab 1200 mg iv on Day 2 of every cycle.	Non-randomized, phase Ib/II, Multicenter, single arm, open-label study	*n* = 60	[120]
R/R AML patients FLT3+	Phase I: establishing the right dose for gilteritinib Phase II: gilteritinib + atezolizumab	Non-randomized, phase I/II, open-label study	*n* = 61	[121]
Relapsed AML/MDS/ALL after allo-HSCT	pembrolizumab 200 mg iv every 3 weeks	Non-randomized, open-label, phase IB study	*n* = 20	[122]
Untreated AML, unfit for intensive chemotherapy	decitabine 20 mg/m^2^ iv day 1–5, every 28 days and avelumab was given at 10 mg/kg iv day 1, every 14 days	Non-randomized, single arm, open label phase I study	*n* = 7	[123]
R/R AML	azacitidine sc/iv days 1–7 or on days 1–5 and 8–9 + avelumab iv days 1 and 14 for 4 courses or until CR and on day 1 for subsequent courses.	Non-randomized, open-label phase Ib/II study	*n* = 19	[124]
MDS patients ≥ 18 years with IPSS-R intermediate, high, and very high or AML patients ≥ 65 years ineligible for intense chemotherapy	azacitidine 75 mg/m^2^ sc, days 1–7 and durvalumab 1500 mg iv on Day 1 every four weeks vs. azacitidine alone	Randomized, open-label, international, multicenter, phase II study	*n* = 213	[125]
R/R AML	pembrolizumab iv 200 mg, day 1 of every three-week cycle + decitabine 20 mg/m^2^, days 8–12 and 15–19	Single-arm open-label, phase I/II study	*n* = 10	[84]
R/R AML patients and newly diagnosed elderly (≥65 Years) AML patients	azacitidine 75 mg/m^2^ iv/sc on days 1–7 every 28 days + pembrolizumab 200 mg iv every 3 weeks starting on day 8 of cycle 1	Multicenter, nonrandomized, open-label phase II study	*n* = 40	[43]
≥60 years AML patients ineligible/refuse intensive chemotherapy	azacitadine iv/sc days 1–7 and venetoclax po days 1–28 of cycle 1 and days 21–28 vs. pembrolizumab iv day 8 cycle 1 and every 3 weeks in cycle 2–6 + azacitadine iv/sc days 1–7 + venetoclax po days 1–28 of cycle 1 and days 21–28 of subsequent cycles.	Randomized phase II, open-label trial	*n* = 76	[88]
≥60 years AML patients in CR not eligible for HSCT	pembrolizumab 200 mg iv once every three weeks	Non-randomized, open-label, phase II trial	*n* = 40	[126]
18–70 years R/R AML patients	Age-adjusted HiDAC followed by pembrolizumab 200 mg iv on day 14 in R/R AML patients	Non-randomized, open-label, phase II trial	*n* = 37	[127]
Newly-diagnosed AML patients	Induction phase: 3 + 7 + pembrolizumab (day 8) vs. 3 + 7. Consolidation phase: HiDAC + pembrolizumab vs. HiDAC. Maintenance phase: pembrolizumab every 3 weeks for up to 2 years	Randomized phase II, open-label trial	*n* = 124	[89]
R/R AML patients or newly diagnosed AML patients not suitable for high-dose chemotherapy or HR MDS or newly diagnosed MDS	AML: pembrolizumab iv days 1 and 22 and decitabine iv days 1–10 MDS: Pembrolizumab iv days 1 and 22 and decitabine on days 1–5.	Non-randomized, open-label, phase Ib trial	*n* = 54	[128]
NPM1 mutated AML patients in CR or MRD positivity or patients not eligible for high-dose chemotherapy or HSCT	pembrolizumab 200 mg iv + azacitidine 75 mg/m^2^ sc	Non-randomized, open-label, phase II trial	*n* = 28	[129]
HR AML (18–78 years)	fludarabine + melphalan+ Autologous HSCT followed by pembrolizumab on day +1	Non-randomized, open-label, phase II trial	*n* = 20	[130]
AML/MDS/cHL, B cell NHL relapsed after alloHSCT	pembrolizumab 200 mg iv every 3 weeks	Non-randomized, open-label, phase I pilot study	*n* = 26	[131]

RFS—Relapse Free Survival, MTD—Maximum Tolerated Dose, MRD-CR—minimal/measurable residual disease negativity and complete remission, ALL—acute lymphoblastic leukemia, DOR—duration of response, DFS—Disease-Free Survival, bid—bis in die, CRp—complete response with incomplete platelet recovery, MPN—myeloproliferative neoplasm, CML—chronic myeloid leukemia, ADA—anti-drug-antibodies, HiDAC—high dose Cytarabine.

**Table 2 pharmaceuticals-14-00288-t002:** Adverse events after PD-1, PD-L1 blockade.

Affected Organ/System	Adverse Event	Symptoms	References
Lung	Pneumonitis	asymptomatic, cough, dyspnea, chest pain, wheezing	[132,133]
	Sarcoidosis	asymptomatic, cough, dyspnea	
Gastrointestinal	Colitis	diarrhea, bloody stools, abdominal discomfort or pain,	[132,134]
	Esophagitis	anorexia, nausea loss of appetite, abdominal pain, nausea, vomiting	
	Gastritis		
	Mucositis		
	Pancreatitis	Fever, nausea, vomiting, abdominal pain with irradiation in the back	[135]
Liver	Hepatitis	asymptomatic, fever, nausea, vomiting	[134]
Skin	Skin Rash		[136]
	Pruritus		
	Vitiligo		
Endocrine	Hypophysitis	fatigue, headache, nausea, postural hypotension, anorexia, tachycardia	[132,134]
	Hypothyroidism	asymptomatic, fatigue, constipation, bradycardia, cold intolerance	[132,137]
	Hypertiroidism	tachycardia, tremor	
	Diabetes mellitus type I	Asymptomatic, polyuria, polydipsia	[132,137]
Ocular	Uveitis	eye redness and pain, decreased vision	[138]
Neurologic	Meningitis, encephalitis, Guillain Barre syndrome, myastenia gravis, polyradiculitis,	nausea, fatigue, headache, blurred vision, dysesthesia, fever, hallucinations, confusion, muscle weakness, tetraplegia, paraplegia	[139]
Cardiac	Myocarditis, pericarditis hypertension, arrhythmias, myocardial infarction	palpitations, dyspnea, chest pain, fatigue	[134,140]
Hematological	Aplastic anemia, hemolytic anemia, immune thrombocytopenia	fatigue, bleeding, infections	[132]
Rheumatologic	Vasculitis, Sicca syndrome, polymiositis, systemic lupus erythematosus	mialgia, joint swelling and pain, dryness of mouth and eye	[141]

## Data Availability

Data sharing not applicable.

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
