# Peer review of "Is There a Place for PD-1-PD-L Blockade in Acute Myeloid Leukemia?"

_pharmaceuticals, 2021, doi:10.3390/ph14040288_

Round 1

Reviewer 1 Report

The paper “Is there a place for PD-1-PD-L blockade in acute myeloid leukemia?” by Jimbu and coworkers are submitted for consideration for publication in Pharmaceuticals. In the paper they describe the use of a programmed cell death protein 1 (PD-1) inhibitors for treatment of AML. They discuss reasons why reasons why checkpoint inhibitors are not very effective in AML therapy such as: the characteristics of the disease (systemic, rapid progressive, high tumor burden disease), low mutational burden and dysregulation of the immune system. Although they conclude therapies could be very helpful in achieving Minimal/ Measurable Residual Disease (MRD) negativity or prior or after transplant, leading to higher graft versus disease effect.

Although, some interesting aspects are discussed, and some part of the manuscript is nicely structured. I have some major concerns which should be addressed before a potential publication.

1, A figure demonstrating the mechanisms of PD-1PD-L blockade in AML would significantly improve the manuscripts and strengthen the impact of the manuscript for the readers.

  1. A table summarizing the amin PD-1-PD-L inhibitors and the pharmacodynamic and pharmacokinetic properties would be of highly interest.
  2. Table 1 is interesting, although should be compromised, as it is way too large.
  3. Discussion of mutation profiles and treatment of other cancers take into too much space. Should be compromised.
  4. What is the actual authors opinion on PD-1-PD-L blockade in AML? This very unsure for me after reading this paper. This should be concretized.
  5. The conclusion is far too speculative and bring in some new moments as not discussed int major paper. The use of PD-1-PD-L to ercitite MRD in AML, are not supported by literature, and although interesting, should be discussed in and own paragraph if the authors believe this is interesting/future potential for PD-1-PD-L blockade. The same is through for the last sentence regarding modulation of graft versus leukemia effect in the post transplantation setting. Conclusion must be significantly altered.

Author Response

Dear Reviewer,

Thank you for your time and for your extensive review. We appreciate the time and effort  that you have dedicated to providing such a valuable feedback on our manuscript . We have been able to assimilate to our paper most of your comments, which were extremely helpful to improve our work. Further, you will find a point by point response to your comments.

Comment 1: we designed a figure demonstrating the mechanisms of action of checkpoint inhibition in AML and we hope our vision will meet your expectation.

Comment2: the pharmacokinetics and pharmacodynamics of PD-1/PD-L1 does not make the object of this review, but it would be interesting to research this subject in another paper

Comment 3: we modified the table according to your suggestions.

Comment 4: as suggested, we comprised the chapter approaching PD-1/PD-L1 inhibitors in non-hematologic cancers.

Comment 5,6: we entirely revised our conclusions in a more assertive way.

Reviewer 2 Report

This review manuscript summarizes the main application of   check point inhibitors on AML disease and report the principal results on clinical trials results.

The manuscript seems well written but in my opinion should be improved in some aspects (please see the comments).

Minor revision:

1) The Authours should include a chapter describe the field of developing computational biomarker for check point inibithors and the  principal results associated with AML if is possible.

2) The Authours should include a chapter on the toxicity on check point inbithors if is possible.

Author Response

Dear Reviewer,

Thank you for investing your time into reading and revising our manuscript. Your insightful comments helped us to improve our review. We have carefully reviewed the comments and have revised the manuscript accordingly. Please, see below a point by point response to your comments.

Comment 1: The subject of computational biomarkers does not make the object of our manuscript. As it is such an interesting and vast field, we believe that it deserves to be approached in a different review paper and we are currently working on it.

Comment 2: As requested, we designed a table and discussed the toxicities of PD-1/PD-L1 inhibitors.

Round 2

Reviewer 1 Report

The manuscript is significantly improved, and my major concerns adressed. I will recomand publication of this paper.